# A Model to Measure U-Learning in Virtual Higher Education: U-CLX

**Gabriel M. Ramírez Villegas [1]**, **César A. Collazos [2]** and **Jaime Díaz [3,*]**

1　Facultad de Ingenierías, Universidad de Medellín, Medellín 050010, Colombia
2　Departamento de Sistemas, Universidad del Cauca, Popayán 190003, Colombia
3　Departamento de Ciencias de la Computación e Informática, Universidad de La Frontera, Temuco 4780000, La Araucanía, Chile
*　Correspondence: jaimeignacio.diaz@ufrontera.cl

**Abstract:** Ubiquitous learning is an evolution of educational learning processes that implements the concept of ubiquity. That is to say, it is found at all times and in all places. This article summarizes our previous works and proposes an alternative to answer our main research question: how can we develop a U-Learning model that integrates connective learning and xAPI user experiences? This paper presents the U-Learning Model Supported by Learning Experiences and Connective Learning for virtual higher education (U-CLX Model) to measure U-Learning in virtual institutions. The U-CLX Model measures ubiquitous learning in four dimensions: time, place, medium, and context. To develop the model, we proposed a theoretical and technological framework, a definition of the U-Learning concept, a unit of measurement for ubiquitous learning (UbiquoL), and a description of the measurement process. We validated the proposal by thematic specialists and applied the instrument in two universities. The model aims to assess the level of ubiquitous learning in virtual higher education institutions and to suggest how these institutions can improve within their current operations.

**Keywords:** ubiquitous learning; virtual higher education; information technology; learning model; interactive techniques

## 1. Introduction

Education has evolved to apply technology in the learning process. One of the evolutions of the learning process is Ubiquitous Learning, or U-Learning, a concept associated with ubiquity. U-Learning refers to students learning anywhere, at any time, in different contexts, and using other media [1]. The learning process occurs without the physical limitations of time, context, and place. U-Learning is applied in any real, virtual, augmented, diverse, or mixed context, using different technologies [2].

Currently, different technologies are applied in higher education and virtual education to support learning processes, such as Electronic Learning (E-Learning), Game Learning (G-Learning), Mobile Learning (M-Learning), and Ubiquitous Learning (U-Learning). All have become essential supports for current learning processes. Education without technology is not conceivable in the present or future [3–5].

In this current work [2] iteration, we propose a U-Learning model that measures ubiquitous learning in virtual higher education institutions. We look to solve the problem of how we can measure ubiquitous learning in virtual higher education institutions.

The importance of the proposed model lies in the possibility of measuring the U-Learning in this kind of institution, which manages educational and teaching–learning processes through education and communication technologies. In this type of institution, online education is not an alternative; instead, it is the basis for daily interaction between students and their teachers. The current education needs to have the possibility of making unique and personalized educational processes.

We reviewed other models and approaches to obtain different contexts and appreciation in order to achieve this. We carried out two different literature reviews and summarized the findings. Our proposal measures the level of ubiquitous learning achieved by virtual higher education institutions (U-CLX Model), considering four dimensions: time, place, medium, and context. We propose a method for measuring U-Learning using the volume equation of a hypersphere in 4 dimensions (R4).

For its validation, we worked with higher education specialists, IT researchers, and mathematical professionals who supported the proposal. Finally, we presented the application in two formal institutions of virtual higher education.

The paper describes the conceptual mathematical basis, the validation, and application of the model, and the results obtained are structured as follows: Section 2 provides the basic methodology of the theoretical proposal; Section 3 provides the fundamental concepts; Section 4 describes the general model; Section 5 discusses the model evaluation environment, the general process, self-assessment, and the indicators of the U-CLX Model. Finally, Section 6 discusses the proposal's validation and results, and Section 7 exposes conclusions and future work.

## 2. Materials and Methods

To develop the model, we performed two literature reviews following the methodology proposed by Kitchenham [6]. Initially, we searched for information about all learning models and methodologies that included ICTs [7,8]. We proposed a new and more specific review, including U-Learning, learning experiences, the xAPI standard, and connective learning [9].

For the execution process, we followed the following steps: (1) we proposed the research questions; (2) presented keywords for searches in the English, Portuguese, and Spanish languages; (3) defined the objective databases; (4) defined the inclusion and exclusion criteria as well as time intervals, (5) generated search strings for each of the databases, (6) executed the search, (7) performed a screening review of the papers and (8) selected and evaluated the final articles.

### 2.1. First Iteration of the Literature Review

The purpose of this review was determined by the current trends in information technologies applied to education and the need to know the current educational models, methodologies, and methods to be used in the learning process [7]. The research questions posed in this iteration was: RQ1: *Which models have integrated educational methodologies and new ICTs?* RQ2: *What methodologies and ICT have been integrated to generate models?*

The search for seven keywords in English, Spanish, and Portuguese was defined: Methodology, Model, Learning, ICT, Integrations, Education, and Pedagogy. The search for seven keywords in English, Spanish, and Portuguese for each language was defined: Methodology, Model, Learning, ICT, Integrations, Education, and Pedagogy. The search was carried out in six databases: IEEE Xplore, SCOPUS, Science Direct, ACM, Web of Science, and Google Scholar.

The inclusion and exclusion criteria for the systematic review were defined according to the topics project and the research questions to perform the searches. These criteria seek to refine previous studies and the proposal [7]:

- Articles published from 2013–2019;
- Articles published in congresses, conferences, journals, and book chapters;
- Articles are written in English, Portuguese, or Spanish;
- Articles related to higher education, virtual education, models and methodologies integrated with ICT;
- The exclusion criteria were:
- Documents not available for download;
- Articles in languages other than English, Spanish, or Portuguese;

- Articles that do not focus on integrating educational methodologies with information and communication technologies;
- Gray literature.

The result of the general search query string is as follows:

*((("methodolog*" OR "methodological") OR ("model*"))) AND ("integrat*") AND ("educat*" OR "learn*" OR "pedagogical") AND ("ICT" OR ("information" AND "communications". AND "technology")) AND (publication year > 2013)).*

From the total of 919 articles found, we finally evaluated 129. The documents that met the inclusion criteria and were appropriate to answer the RQ comprised 14.04%. We rejected 85.96%, as they did not meet the inclusion criteria. The general summary of the documents accepted and rejected by the databases is detailed in Figure 1.

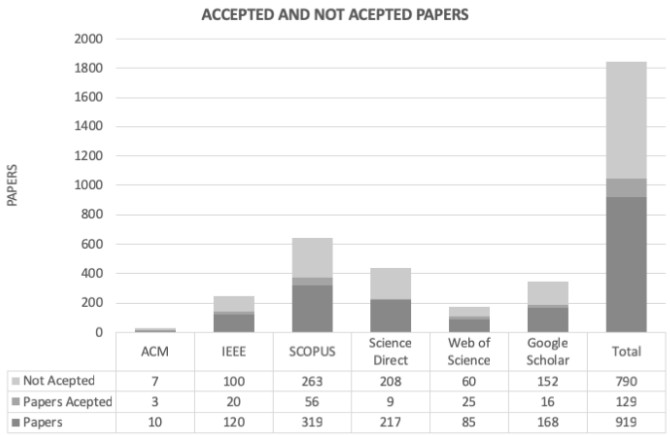

**Figure 1.** Accepted ratio from the DDBB—First Iteration [2].

RQ1: The most relevant models and related found in the systematic review are: (a) TPACK model; (b) mobile learning and technology acceptance TAM model; (c) maturity model for assessing ICT use; (d) the eSG project: a blended learning model for teaching entrepreneurship through serious games, development of an instructional model based on learning connectivism; (e) e-Inclusion modeling for blended analysis, social and ecological model for ICT integration; (f) ubiquitous learning: modeling and simulation of teaching with technology; (g) a cloud model for effective e-learning, collaborative learning based on web 2.0-based collaborative learning, a study circle model, hierarchical model for E-Learning implementation challenges using AHP; and (h) a conceptual framework for enhancing motivation in an open learning model learning environment, the flipped classroom model at the university.

The model approaches found in the review allow an understanding of the need to create models that integrate new information and communication technologies with educational processes. The most important answer is the different ways to integrate technological elements in education, designing models that respond to specific problems.

The need to implement technologies in education is undeniable. In this sense, we found exciting models focused on learning using ICT as a medium. However, the models that integrate educational strategies and ICT only work in a specific context and provide little information on the integration process.

For example, The TPACK model prepares and evaluates lesson plans. It is an experience with pre-service teachers using social networks and digital resources.

The papers in the review propose further studies and applications covering new elements to obtain better results. This allows the proposed models to evolve, as well as the possibility of conducting a recent systematic review focused on U-Learning and the relationships with technological and pedagogical elements.

RQ2: The essential methodologies found in the systematic review are: (a) a methodology of developing additional content in an adaptive agent-based e-learning environment; (b) the evaluation of ICT integration in higher education: Foundation for a Methodology; (c) parallel virtual urban workshop: a 'reasonable cost' methodology for academic internationalization in problem-solving. Graduate-oriented subjects and web 2.0 tools for role-playing methodology in an interdisciplinary undergraduate environment.

The papers presented were the most relevant in terms of methodologies; the significant contribution of the articles found concerning the research question focuses on the approach of virtual resources in educational areas where agents and environments adaptable to educational media can be developed. The evaluation of integrations concerns the effectiveness of the educational process and assessment, as well as problem-solving in interdisciplinary environments. However, the systematic review found few works on implementing ICT and educational processes.

In conclusion, according to the results obtained in the review, the development of a new study focused on U-Learning, Connective Learning, and Learning Experiences is considered based on the integration model proposed in the research. The latest systematic review will search for the topics of intelligent learning and deep learning for the pedagogical component within the framework of connective learning theory.

### 2.2. Second Iteration of the Literature Review

The new systematic review process followed the same steps as the first. The purpose of the second systematic review focused on U-Learning, connective learning, and the xAPI user experience standard to concentrate on discovering the conceptualization, process, and operation of U-Learning. RQ3: *How to develop a U-Learning model that integrates connective learning and xAPI user experiences?*

A search for six keywords in English, Spanish, and Portuguese included the results of the database searches in these languages: xAPI, User Experience API, Tin Can API, U-Learning, Ubiquitous Learning, and Connective Learning. As in the previous review, we searched six databases, and the inclusion and exclusion criteria were the same.

We defined a general query, but no results were found with the three keywords. We then described a string with two keywords, and no results were found either, which is why we decided to perform searches by words. Word String Term 1: xAPI, TIN CAN API, User Experience API; Word String Term 2: U-Learning, Ubiquitous Learning; Word String Term 3: Connective Learning.

The graph shows that the search by terms generated 824 papers, of which 767 were rejected because they did not meet the established inclusion criteria. We finally selected 57 articles. Figure 2 shows the papers accepted in each database in detail.

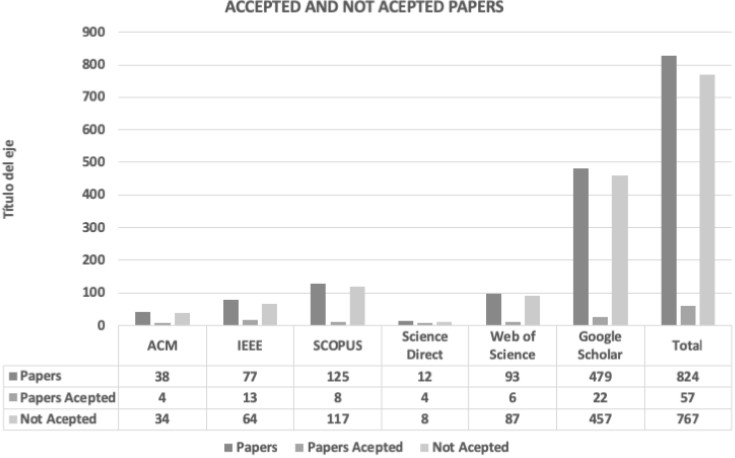

**Figure 2.** Accepted and rejected papers, second systematic revision [2].

RQ3: The works found the relationships between U-Learning and different concepts and technologies. We found some ontologies referred to as U-Learning and measurement models implemented in other institutions [9].

The research was designed to implement the xAPI standard with an LMS and to mix pedagogical strategies with U-Learning.

Connective learning has been proposed as one of the theoretical bases of U-Learning. Philosophical, ethical, and legal debates have been raised regarding managing data generated with U-Learning and the xAPI standard.

The book "The future of Ubiquitous Learning" [10] exposes the need to update pedagogies and apply new technologies in education to implement U-Learning. We found no U-Learning models based on connective learning and the xAPI user experience standard. In total, 58% of the works that directly study on U-Learning issues suggest the need to investigate and continue developing models, methodologies, and strategies. However, various authors have exposed the same problem [8,10–12].

With these results, it is possible to conclude that it is feasible to develop a U-Learning model based on learning experiences and connective learning. As observed in most of the papers reviewed, many models implement ICT in education.

It is necessary to propose conceptualizing and characterizing the concepts of U-Learning and connective learning as well as learning experience to develop the model.

### 2.3. Discussion of the Proposal's Scope after the Literature Review

As stated earlier in the paper, according to the systematic reviews conducted, we did not find U-Learning models that allowed the measurement of ubiquitous learning in virtual higher education institutions. However, there are models related to U-Learning that evaluated other measurements, allowing further discussion on this topic.

A study in Indonesia evaluated this country's readiness for and success of ubiquitous learning. Insights from implementing a pilot project raise the opportunity to address affordability, accessibility, and quality issues in the higher education sector. However, its wide application still needs to be improved within the instructional paradigm. The proposal aimed to understand the readiness for and success of e-learning implementation in Indonesia, and to assess the factors influencing the status based on stakeholder perspectives [13].

Laisema [14] proposed a collaborative learning model and problem-solving in a ubiquitous learning environment. The model aims to develop creative thinking and collaborative skills, and consists of four components: principles, objectives, instructional process, and assessment. In the end, the model performs a learning assessment measuring creative thinking.

Jung [15] presented a model that incorporates the characteristics of ubiquity (ubiquity, context personalization, interactivity, self-directed learning, and perceived enjoyment) and learner characteristics (innovativeness, learning motivation, and computer self-efficacy) and their impact on English language learners' satisfaction. The study evaluates the effects of satisfaction on expectations in the context of English language learning and employs structural equation modeling (SEM) to test the hypotheses. The study indicates that all ubiquity and two learner characteristics variables (innovativeness and computer self-efficacy) had significant effects on satisfaction with U-learning, and satisfaction positively affected expectations.

Yun et al. [16] proposed a model for reliability analysis of systems with multiple failure modes using a ubiquitous learning function. The model faces the challenge of accuracy and efficiency, enhancing AK-SYS using a refined U-Learning function that updates the Kriging metamodel.

Caytiles [17] proposed U-Learning Community as an interactive social learning model based on wireless sensor networks. It is a ubiquitous learning environment system based on the concepts of ubiquitous computing technology that allows learning to take place anywhere and anytime. The U-learning model is a web-based e-learning system that could

enable learners to acquire knowledge and skills through interaction between them and the ubiquitous learning environment. Communication between devices and computers embedded in the environment allows learners to learn in an environment of their interest while on the move, linking them to their learning environment.

Later, Caytiles, and Kim [18] proposed an interactive social learning model of U-Learning that allows for a ubiquitous learning environment. This environment is described as an environment that supports student learning using digital media in geographically distributed environments. The U-learning model is a web-based e-learning system that could enable learners to acquire knowledge and skills through interaction between them and the ubiquitous learning environment.

Durán et al. [19] proposed an ontological model for the personalization of U-learning applications. The authors presented a broad conceptual framework that models the semantic context of ubiquitous learning applications and explains the reasoning that can be used from this model to infer new knowledge for personalization purposes. This framework can allow a software developer to take appropriate action concerning building a ubiquitous learning application with all the classes, relations, and decision reasoning rules. In addition, it will enable intelligent agents to reason about contextual information, and be able to provide personalization services to ubiquitous applications.

Xiao et al. [20] designed an augmented reality-based learning system applied in the U-Learning environment. Augmented Reality (AR) can be recognized as a critical technology used in the U-learning environment to enhance the learning effect and improve the learning experience. The model integrates U-learning and a learning system based on Augmented Reality technology called "Starry Sky Exploration—Eight Planets of the Solar System".

Xiao et al. [21] evaluated the application of learning analytics to estimate the learning effect using a mobile learning support system in a U-Learning environment. The authors aimed to design a practical Learning Analytics (LA) model and essential analytics indicator to apply this LA model in order to evaluate the learning effect by using a mobile learning support system. It is an augmented reality (AR) learning APP in a U-learning environment. An in-depth learning analysis is conducted for the learner behavior data in three dimensions: access behavior, interactive behavior, and learning effect. From the data analysis and evaluation, the designed learning analysis model and analysis indicators can represent the behavioral characteristics of learners in the U-learning environment and provide a relatively systematic and comprehensive data analysis.

Chen at al. [22] proposed a context-adapted teacher training model in a ubiquitous learning environment. The model provides teachers of different subjects with adaptive and personalized learning content in an online learning environment, implements intra- and intergroup collaboration to facilitate knowledge construction and in-depth study, and promotes reflection with the help of supervising teacher's review and summarization.

Boudabous et al. [23] described an agent model for the U-learning system AMuL. It uses a multi-agent system (MAS) to facilitate access to target information anywhere and everywhere. The authors proposed an agent prototype for the UL system to facilitate complicated learning tasks. The objective is to demonstrate the combination of Agent-Oriented Software Engineering (AOSE) methodologies used throughout all phases of software development as well as Model Driven Engineering (MDE).

Cárdenas-Robledo [24] proposed a holistic model of self-regulated learning. Technology-enhanced learning (TEL) represents an expert and intelligent paradigm that uses technological affordances to facilitate learners' acquisition of domain knowledge (CD).

Moreno-López et al. [25] described a learning model for education and training processes with the support of TV Everywhere platforms, using technological advances and digital convergence such as Netflix, which allows users to watch TV and video without time or place restrictions. These advances can be applied to education and training processes to enable ubiquitous learning (U-Learning). They are explicitly using applications that allow access to the TV regardless of location and device. To contribute to this and other

challenges in education, the objective of this model is to implement U-learning involving TV/video platforms supported in the cloud.

Restrepo et al. [26] proposed benchmarks to evaluate the level of the ubiquity of a higher education institution. The model is built on three dimensions: Technology, Learning, and Management, which are evaluated through the identification of categories, properties, and their associated metrics and indicators to determine the levels of ubiquity in a higher education institution.

Naatonis et al. [27] evaluated perspectives on the philosophy of *progressivism education* in ubiquitous learning models, which leverage digital content, physical environments, mobile devices, ubiquitous components, and wireless communication. The authors declared that the conception of ubiquitous learning is in line with the philosophical view of progressivism, which believes that education should consistently innovate or change according to the changing times and science and technology. In another sense, the philosophy of progressivism is a school of modern educational philosophy that wants a change in the application of education to be more advanced. The study aims to analyze U-learning from the perspective of the philosophy of progressivism in order to evaluate the relationship the compatibility between the two concepts in education.

Once the different models' research, studies, and proposals on U-Learning are presented, we can declare that we did not find models that measure U-Learning in virtual higher education institutions, considering the components of ubiquitous learning, computing and the dimensions of time, environment, place, and context. The u-CLX model is relevant and contributes to the development of U-Learning in virtual higher education institutions.

## 3. Background

This section presents the main concepts for constructing the model from the conceptual and mathematical perspectives that support the proposal for measuring U-Learning, followed by different related concepts with identification of their respective authors.

### 3.1. Ubiquitous Computing

The first known author to have worked on the subject of ubiquitous computing is Weiser [28], who initially proposed this term. He proposed the daily application of computing in all areas of people's lives, using technology to minimize and mimic the functions of objects of everyday use, in a way that is invisible and imperceptible to users. However, according to [10], U-Learning raises the omnipresent problem of maintaining the continuity of learning in ubiquitous processes.

### 3.2. Connectivism

According to Siemens [29], connectivism integrates the theories of chaos, networks, complexity, and self-organization, which are new ways of learning in today's world. Learning is a process that occurs within diffuse, complex, and changing environments and elements. It is defined as the development of relevant knowledge inside and outside people, in any time, space, or context. It focuses on connecting specialized information sets, and its connections generate new knowledge.

### 3.3. U-Learning

Schilit et al. [30] defined U-learning over time and proposed the relationship between ubiquitous learning and context, which makes it vital to obtain all the data related to the context, such as localization and ubiquity. Subsequently, Hummel and Hlavacs [31] proposed that services and web platforms promote interaction, as well as the availability and use of mobile devices, for ubiquitous learning purposes at any time and place, thus increasing U-learning coverage. Finally, Bomsdorf [32] proposed the adaptation of learning spaces supported by ubiquitous learning.

S. Yang et al. [33] studied the context of learning environments to promote collaborative learning between peers. The result of this approach was to increase the students'

understanding. Furthermore, when the students' preferences in terms of surroundings, services, and technologies were respected, ubiquitous learning was promoted. In Hwang, Tsai, and S. Yang [34], ubiquitous learning represented learning at any place and time, since the learning environment allowed students to access content from anywhere and at any time. Graf, Yang, Liu, et al. [35] defined U-Learning as adaptable to students, in that it can be conducted at any time and place, allowing students to adapt learning material and carry out personalized activities.

According to some authors, such as Yahya et al. [3], U-Learning is the evolution of learning through technology and its application in education; U-Learning is, thus, the evolution of E-Learning, M-Learning, T-Learning, B-Learning, G-Learning, etc. The inclusion of technology in each type of learning results in a higher baseline and further improvement in ubiquitous learning. Yamamoto et al. [36] proposed the evolution of U-Learning and used a Cartesian plane to relate trends in education with information and communication technologies (ICT), and to display hardware, software, and other technologies. Rinaldi [37] mentioned that U-Learning is all of the above, with the addition of Web 2.0 and other forms of ICT learning.

### 3.4. Learning Experience

Learning experience refers to the interaction of any person or people with courses, programs, objects, technologies, or any other experience in which learning occurs. In other words, everything that generates new knowledge is a learning experience; this can happen in any place, time, context, and medium, or a combination of all these [38]. It can be developed in traditional or non-traditional education environments, with the participation of teachers, peers, colleagues, or strangers, through personal relationships or interactions with other people. It can also refer to learning involving tangible or intangible objects, using games, video games, interactive software, applications, web services, and robots [38].

### 3.5. Seamless Learning

Seamless and continuous ubiquitous learning, proposed by Wong and Looi [39], refers to the seamless integration of learning experiences in different formal and informal dimensions and contexts, individually and socially, in the physical and virtual world. Sampson et al. [40] mentioned context-sensitive knowledge and personalization of learning in formal and informal ubiquitous learning; the use of cloud computing, mobile computing, location-based services, serious games, and ubiquitous computing are the basis for developing U-Learning.

Each individual learns differently and at a different pace, creating unique learning experiences [41]. In conclusion, learning experiences are all the interactions between people and natural or abstract objects, which allow the development of a learning process in order to generate new knowledge [42].

### 3.6. U-Learning Measurement Models

Currently, there are models, methodologies, and frameworks that have been tested to measure the level of the ubiquitous learning of educational institutions, such as the TAG Model [43]. Learning processes, technologies, and other areas have been examined by Bomsdorf [32], Kwon [44], and Poslad [45], but so far, no model exists for measuring U-Learning in virtual higher education.

## 4. The U-CLX Model Proposal

The U-Learning model, supported by connective learning and learning experiences for virtual higher education, is a U-CLX Model framed in a ubiquitous learning ecosystem. The two components, Pedagogical and Technological, have four dimensions, and each component has six elements [2].

The main purpose of the U-CLX Model is to measure ubiquitous learning (U-Learning) in virtual higher education institutions. The U-CLX Model measures ubiquitous learning

at any time, place, medium, or context. Considering that ubiquitous learning is related to people, we concluded that ubiquity is inherent to people and their learning processes.

Ubiquitous Learning is the basis of the Pedagogical component, and Ubiquitous Computing is the basis of the Technological component; each component has elements by which the level of ubiquitous learning is measured. The relationship between the components and their elements generates the possibility of measuring the level of ubiquitous learning in a virtual higher education context. The measurement of ubiquitous learning by the U-CLX Model has a mathematical basis, found in the volume equation of a four-dimensional hypersphere. The U-CLX Model has four dimensions, two components, and twelve elements in total [2] (see Figure 3).

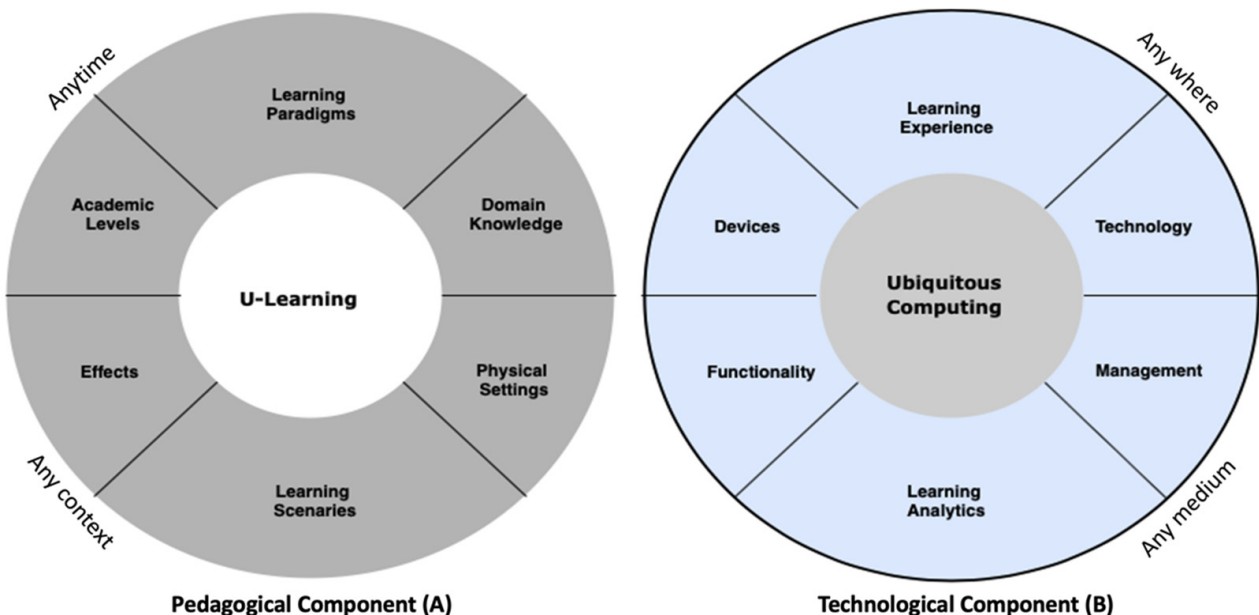

**Figure 3.** U-CLX Model Overview [2].

U-Learning is the basis of the Pedagogical Component (A). The elements of this component are Learning Paradigms, Domain Knowledge, Physical Settings, Learning Sceneries, Effects, and Academic Levels. All of these elements are related to ubiquitous learning processes.

Ubiquitous Computing is the basis of the Technological Component (B). The elements of this component are Learning Experience, Technology, Management, Learning Analytics, Functionality, and Devices.

Below, we give a specific and detailed view of the U-CLX Model, with the dimensions, components, elements, learning processes, and people. The model's components are composed of elements, which are the definitions and conceptual characteristics needed to measure ubiquitous learning. The model is based on U-Learning taxonomy and patterns [1]. Within the ecosystem, people and learning processes will measure the level of ubiquitous learning in the four dimensions of the U-CLX Model (see Figure 4).

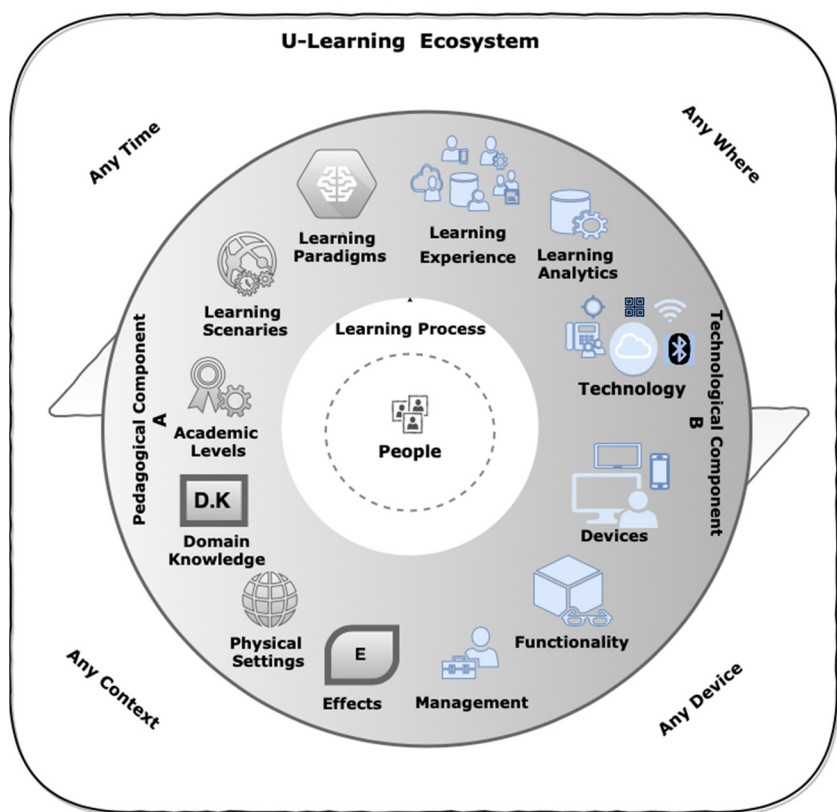

**Figure 4.** U-CLX Model Specific View [2].

*4.1. Conceptual Basis of the U-CLX Model*

In this section, we will present the basic theoretical aspects of our proposal. These are detailed below: model metrics, theoretical dimensions, learning experiences and the entities involved, and pedagogical and technological components.

4.1.1. U-CLX Model Metrics

Measurement processes are defined as the assignment of numbers, symbols, or values to the characteristics of objects in the natural, virtual, or intangible world. This gives us a clear and concise description without generating ambiguity or confusion about the described object [46]. The primary metrics of our proposal are described below.

- Attribute: A measurable characteristic of an object or entity.
- Measurement: The process of assigning symbols or numbers to the attributes of an object to describe them according to pre-established rules [47].
- Measure: The assignment of a number or symbol from a measurement process to an object to characterize an object [47].
- Metric: A quantitative measure of the degree to which a system, component, or process possesses an attribute [46].
- Indicator: A value defined by the measurement process according to ranges established in the metrics. An indicator is a variable that can be defined by the results of a measurement [47].

In the case of the U-CLX model, metrics are proposed to measure the level of U-Learning quantitatively related to the dimensions, components, and elements defined, taking into account the general reference of ubiquitous learning defined in the U-CLX model. The proposed implementation creates indicators to measure ubiquitous learning in the U-CLX model.

### 4.1.2. U-Learning in the U-CLX Model

The evolution of U-Learning has allowed us to propose our concept. In the U-CLX Model, we propose U-Learning as a ubiquitous learning ecosystem centered on people, processes, and learning contexts. People's learning experiences in different times, places, and contexts, and using different media, are measured through components and elements such as information and communication technologies, learning analytics, pedagogical elements, theories, paradigms, methodologies, and learning strategies [2].

### 4.1.3. Dimensions

- The dimensions of the U-CLX Model are Time, Place, Medium (Device), and Context. According to the U-CLX Model, people are the center of the model, and their learning processes are where U-Learning is measured. Therefore, the left side of the model represents the Pedagogical component (A), and the right side the Technological component (B). Each component contains the elements for evaluating U-Learning with the U-CLX Model. Table 1 presents dimensions of the model.

**Table 1.** U-CLX Model Dimensions.

| Dimensions of the U-CLX Model | |
|:---:|:---:|
| A<br>N<br>Y | Time<br>Where<br>Device<br>Context |

- Any Time: this dimension refers to the possibility of carrying out learning processes and experiences simultaneously, at any time, or at different times. It is possible to carry out these processes online or offline, and to carry out learning processes continuously at any time without affecting learning.
- Anywhere: this dimension refers to the possibility of carrying out learning processes and experiences in any place or physical space in the natural or virtual world, or a mixture of both. It also refers to the possibility of moving in any direction and continuously developing these learning processes, without being affected by physical or virtual changes of location.
- Any Medium (Device): the medium or device refers to the possibility of using any technological element (computers, servers, smart devices, sensors, networks, cloud technologies, standards, interactions with different technologies, etc.) that allow ubiquitous learning processes and experiences using computing and ubiquitous technologies.
- Any Context: this dimension refers to the possibility of developing learning processes and experiences in any context in the real, virtual, or augmented world, for formal or non-formal education, and by a variety of people. It refers to the learning process defined by each person's unique context.

The dimensions are the model variables with which the level of U-Learning will be measured. Each component has its elements and attributes. The evaluation is performed on these elements. Their respective components are evaluated in the four dimensions model: any place, time, moment, and context to measure ubiquitous learning.

In each dimension, the institution's compliance is measured according to the elements, i.e., the evaluation indicates whether it has a low, medium, or high compliance. This evaluation makes it possible to quantify how an element of the model exists in the four dimensions. By evaluating all elements of the two components in the four dimensions, it is possible to indicate the level of U-Learning of a virtual education institution.

### 4.1.4. People, Roles, Learning Processes, Context, and Learning Experiences

- People are the principal component and central element of the U-CLX Model. The idea of the model is to evaluate levels of ubiquitous learning according to people's learning experiences, as well as the context. There are three classes of participants in the model: Academic Manager, Learning Engineer, and Student.
- Learning processes are the cognitive mechanisms of reception, assimilation, and data analysis that allow the generation of information and production of knowledge, for the purpose of forming intelligence and obtaining wisdom to apply in different situations and contexts. Learning processes are individual and depend on students' characteristics, forms, skills, attitudes, and learning abilities.
- Learning contexts are the scenarios in which the ubiquitous learning processes and experiences occur. Each context is personal and unique, since each person creates their learning with different pedagogical and technological elements, which form the learning context.
- Learning experiences are characterized by the capture of all data generated in the ubiquitous learning process, followed by data analysis to develop information, knowledge, intelligence, and wisdom, thus allowing ubiquitous learning to occur.

### 4.1.5. Pedagogical Component A

- The pedagogical component of the U-CLX Model is the interaction between the different actors in the learning processes and experiences, where communications, knowledge construction, and data generation are necessary for analysis of the experiences of ubiquitous learning. This component includes all the educational elements and is the pedagogical and conceptual basis of the whole model. The elements are Learning Paradigms, Learning Scenarios, Academic Levels, Knowledge Domains, Physical Characteristics, and Effects.
- Learning Paradigms are a set of items that refer to how people acquire and build knowledge through ubiquitous learning processes and experiences. The items that belong to Learning Paradigms are Learning Styles, Educational and Pedagogical Theories, Techniques, Methodologies, and Learning Strategies. The sub-items are Authentic Learning, Research-based Learning, Social Constructivism, Continuous Learning, Self-regulated Learning, Learning by Doing, Learning Theories, Learning Techniques, Connective Learning, etc.
- Learning Scenarios are spaces where ubiquitous learning processes and experiences are carried out. U-Learning learning scenarios can occur in the real and virtual worlds, combining virtual and augmented reality with social interaction, individual work, collaboration, cooperation, and learning networks.
- Academic Levels represent the hierarchy of the structure and organization of education from the bottom up, including the classification of education into formal, informal, and mixed, in which people can carry out ubiquitous learning processes and experiences. The academic levels are primary, secondary, high school, professional, graduate, and lifelong learning. Certain competencies, knowledge, physical and mental skills, attitudes, and states of maturity are recognized, acquired, and accredited in the different educational levels.
- Domain Knowledge is how to provide, deliver, and build knowledge in U-Learning. This includes the development of specific cognitive skills, participation in different learning experiences and processes, generating knowledge in people, and allowing them to use, apply, and develop a theme or set of themes in different contexts or settings.
- Physical Settings are the places where ubiquitous learning processes and experiences take place, providing the possibility of U-Learning anywhere, either indoors (universities, classrooms, laboratories, etc.) or outdoors (campus, gardens, zoos, urban spaces, etc.).

- Effects are the changes and results generated in people by their interaction with ubiquitous learning processes and experiences. Effects refer to the influences, reactions, changes and evolutions, creation, and formation of people in ubiquitous learning experiences. The effects are classified into people's points of view, commitment, motivation, emotions, learning goals, learning competencies, learning results, feelings, meta-cognition, reflections, awareness, regulation, socialization, cognitive load, collaboration, thinking, etc.

4.1.6. Technological Component B

- The technological component of the U-CLX Model is technological support, consisting of technical and technological elements that allow ubiquitous learning processes and experiences to take place. These elements are the basis for the development of ubiquitous learning using the four dimensions of the model. This component includes Learning Experiences, Analytics, Technologies, Devices, Management, and Functionalities.
- Learning Experiences are ubiquitous learning processes in which people have the central role. The activities in the different levels of training include the four dimensions of the model (time, place, medium, and context). On the technical and technological side, learning experiences capture all data generated in the U-Learning processes of people; these data are managed, through analysis, to generate information, knowledge, wisdom, and intelligence.
- Learning Analytics: data are collected, measured, analyzed, presented, and reused in order to obtain information about the data generated by people in the different contexts and interactions of ubiquitous learning processes and experiences.
- Technology: all information and communication technologies by which data can be sent and received can produce ubiquitous learning processes and experiences. This element refers to the hardware and software involved in ubiquitous learning, for example: the identification of QR labels, RFID, GPS global positioning, NFC, Bluetooth, WIFI, SMS, satellite, sensors, beacons, and the software to develop these learning processes, as well as widely used technology such as ubiquitous computing, cloud computing, big data, data analytics, etc.
- Devices: all intelligent technological devices with computing and communication capabilities that can capture and measure data generated in ubiquitous learning processes and experiences, such as smartphones, tablets, laptops, cameras, microphones, televisions, watches, and wearable devices.
- Functionalities: this element refers to the design and development of environments for U-Learning, specifying how the processes and experiences should work in ubiquitous learning, and the interaction between academic staff and students. It provides a description and explanation of ubiquitous learning processes. The functionalities are educational support, delivery of content, time, place, medium, and context, correctly applied to ensure and facilitate the U-Learning process.
- Management administers the technological and pedagogical elements used in ubiquitous learning, through management of the devices, technologies, learning analysis, learning experiences, and functionalities used by people in U-Learning.

*4.2. The Mathematics under the U-CLX Model*

Once the requirements, functions, and theoretical specifications of our model had been defined, we decided to add an individual logical–mathematical behavior. This approach has been reported in specific studies [43], and in our case it was based on computation of the elements proper to ubiquity. For example, some of these elements had a maximum of only three study variables; therefore, they used elements of vector algebra mathematics, such as three-dimensional hyperplanes.

Since the U-CLX model has four dimensions, the model is based on the concepts of the Rn points, lines, vectors, hyperplanes, convex sets, and hyperspheres. In this case, the

calculations are in four dimensions, namely R4, time, where, device, and context. The U-CLX model is based on the TAG Model's mathematical idea [26], which measures the level of ubiquity in institutions in three dimensions: learning, technology, and management.

Because the U-CLX Model has four dimensions, it uses the hypersphere concept in R4. We propose an equation to calculate the volume of a figure in a hypersphere, which will allow us to estimate the level of ubiquitous learning in the U-CLX Model using the volume equation of a hypersphere in four dimensions (R4). In mathematics, an n-sphere is the generalization of a sphere to a Euclidean space of arbitrary dimension.

The volume of the hypersphere with radius *r*, in the space of four dimensions with volume n = 4, is given by the following Equation (1) (Henderson [48] and Cederberg [49]):

$$V_4 = \frac{\pi^2 r^4}{2} \tag{1}$$

Note that the U-CLX Model defines four dimensions, each of which is a plane inside the hypersphere: time (T), place (L), medium (M), and context (C). It is flat in the hypersphere, and, therefore, the four planes of the hypersphere correspond to the four dimensions.

The mathematical description of the model considers the following assumptions: (a) all four variables have the same weight or value, (b) all variables have the same level of development, and (c) all are positive. This assumption is due to the ease of calculating the variables. In addition, no previous study has been conducted to indicate which variable has a greater or lesser weight in the model.

In the U-CLX Model, all points, lines, planes, and hyperplanes are inside the hypersphere in four dimensions or R4, so the calculated volume of the hypersphere will give the measure of Ubiquitous Learning.

The four dimensions of the U-CLX Model are the variables with which it is possible to measure the level of ubiquitous learning through the hypersphere. The result is a 4D volume in a solid sphere with four dimensions. The variables cannot be negative; dimensions in the hypersphere start at points (0,0,0,0); the level of the variables or dimensions begins at 0, and the level is E = 0, as shown in Figure 5.

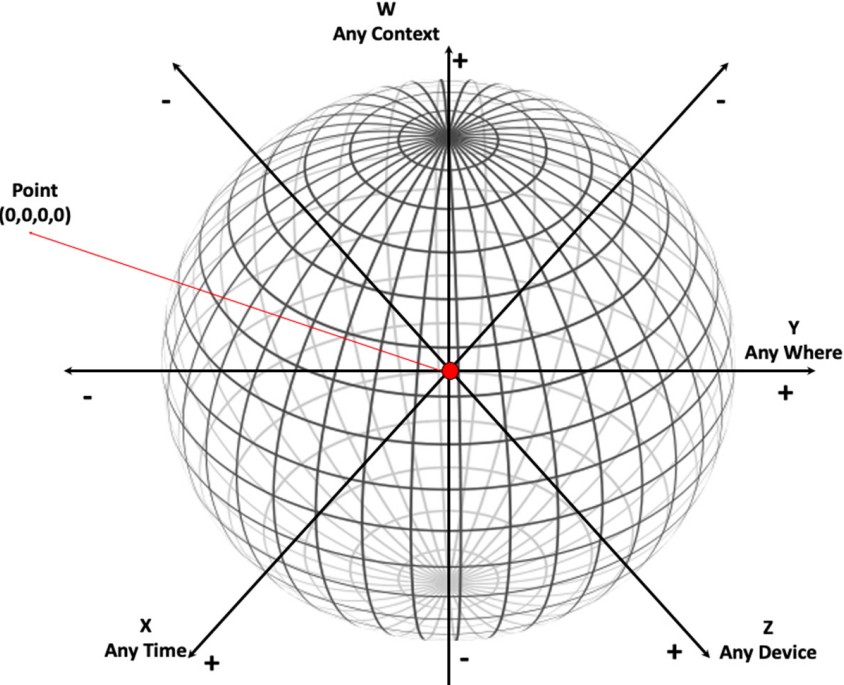

**Figure 5.** Hypersphere in four dimensions [2].

### 4.2.1. U-CLX Model Equation

The U-CLX Model equation for calculating the level of U-Learning is based on the calculation of the volume of a 4-dimensional hypersphere in R4. It measures ubiquitous learning based on the volume of a hypersphere, in which there are four variables with minimum and maximum restrictions. Therefore, the maximum value is the maximum level of ubiquitous learning, and the minimum value is the minimum level of ubiquitous learning.

The mathematical basis of the model is based on the following premises: there are four (4) variables (Time, Place, Medium, and Context), and all the variables have the same weight or value. Therefore, all the variables have the same level of development; they are all positive. The concepts of point, line, segment, and hyperplane are all contained within the hypersphere in the model.

For this reason, the calculations and measurements inside the hypersphere correspond to Ubiquitous Learning, or U-Learning, in the U-CLX Model.

The U-CLX Model's four dimensions for measuring U-Learning are: any time (T), any place or space (P), any device or medium (M), and any context or reality (C), which are mutually related. The dimensions are the variables of the U-CLX model and, at the same time, are the points in the hypersphere:

- $x$: is the Time (T), which is a plane of the form P1 = (v1; 0; 0; 0)
- $y$: is the Place (P), which is a plane of the form P2 = (0; v2; 0; 0)
- $z$: is the Medium (M), which is a plane of the form P3 = (0; 0; v3; 0)
- $w$: is the Context (C), which is a plane of the form P4 = (0; 0; 0; v4)

The points of the four dimensions form coordinates in the hypersphere, and these points can be used to define the radius ($r$) of the hypersphere in R4 (2).

$$r^2 = (x - x_0)^2 + (y - y_0)^2 + (z - z_0)^2 + (w - w_0)^2 \tag{2}$$

The spherical coordinates of the U-CLX Model correspond to

$$x = \rho \, \sin(\varphi) \, \sin(\varnothing) \cos(\varnothing) \tag{3}$$

$$y = \rho \, \sin(\varphi) \, \sin(\varnothing) \sin(\varnothing) \tag{4}$$

$$z = \rho \, \sin(\varphi) \, \cos(\varnothing) \tag{5}$$

$$w = \rho \, \cos(\varphi) \tag{6}$$

If $\rho = 1$, the above expressions correspond to a hypersphere of 4D dimensions.

The U-CLX model calculates the ubiquitous learning level through the volume equation of the hypersphere in R4, and is given by the Equation (1), and

$$r = \sqrt{x^2 + y^2 + z^2 + w^2} \tag{7}$$

$$r^2 = x^2 + y^2 + z^2 + w^2 \tag{8}$$

where

- $x$ is the Time T
- $y$ is the Place P
- $z$ is the Medium M
- $w$ is the Context C
- Whose conditions are:

$$0 \leq x, y, z, w \leq 100 \tag{9}$$

Scaling the U-CLX Model equation, it is essential to assume that the maximum values of the four dimensions (T; P; M; C), i.e., ($x$; $y$; $z$; $w$) = 100. The equation must be scaled in

order to calculate the maximum level of ubiquitous learning or U-Learning; for this reason, the following equations and solutions must be used.

### 4.2.2. U-CLX Pyramid

According to the U-CLX Model, the volume calculation is in the positive part of the dimensions and of the hypersphere. Therefore, the volume calculation is performed only in the positive part of the hypersphere. This means that the positive parts of the dimensions of the hypersphere form a square-based pyramid, as shown in Figure 6.

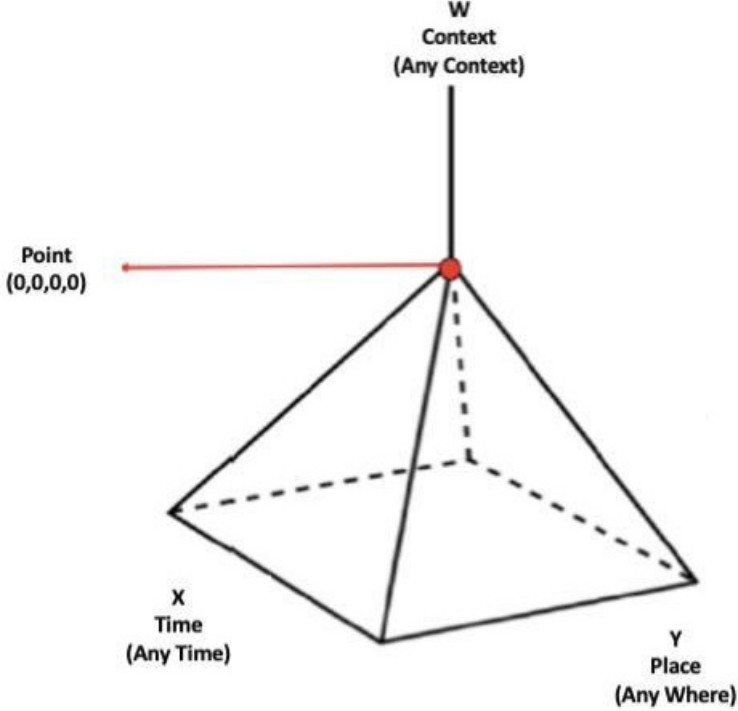

**Figure 6.** U-CLX Model Pyramid [2].

### 4.2.3. Scaling the U-CLX Model Equation

Bearing in mind that the maximum values of the dimensions are $x$; $y$; $z$; $w$ = 100, the equation must be scaled to take the total value of 100 for the maximum level of ubiquitous learning or U-Learning; in order to do this, the following equations and solutions must be used.

$$V_4 = \frac{\pi^2 r^4}{2} = \frac{\pi^2}{2}\left(r^2\right)^2 \quad \text{With} \quad r^2 = x^2 + y^2 + z^2 + w^2$$

$$V_4 = \frac{\pi^2 r^4}{2} = \frac{\pi^2}{2}\left(x^2 + y^2 + z^2 + w^2\right)^2 \tag{10}$$

$x, y, z, w$ has a maximum value of 100.
Where $V_{4_T}$ is the maximum value,
When $x = y = z = w = 100$,
Then, the maximum value of the variables in the equation.

$$V_{4_T} = \frac{\pi}{2}\left(100^2 + 100^2 + 100^2 + 100^2\right)^2 \tag{11}$$

$$V_{4_T} = \frac{\pi}{2}\left(4 \times 100^2\right)^2 \tag{12}$$

$$V_{4_T} = \frac{16\pi}{2}\left(4 \times 100^4\right) = 8\pi\left(100^4\right) \tag{13}$$

$V_{4_T}$ is the maximum value of volume.

Because the volume is the maximum value, $V_{4_s}$, for the maximum values of $x; y; z; w$, V4S must be scaled. The following relation is observed:

$$V_{4_T} = \frac{16\pi}{2}\left(4 \times 100^4\right) = 8\pi\left(100^4\right) \tag{14}$$

Suppose the scaled $V_{4_S}$ is 100 when the maximum total volume $V_4 = V_{4_T} = 8(100)^4$; then the ratio for the new scaled volume is:

$$V_{4_T} = \frac{16\pi}{2}\left(4 \times 100^4\right) = 8\pi\left(100^4\right) \tag{15}$$

$$V_{4_S} = \left(4 \times 10^{-8}\right) \times \left(\frac{\pi}{2}r^4\right) \tag{16}$$

$$V_{4_S} = 6.28 \times (10)^{-8}r^4 \tag{17}$$

where the equation is:

$$r^4 = \left(x^2 + y^2 + z^2 + w^2\right)^2 \tag{18}$$

$V_{4_S}$ is the volume scaled to be equivalent to 100, when

$$x = y = z = w = 100$$

The U-CLX Model is developed in a four-dimensional hypersphere. The model has four dimensions, and each dimension is a variable of the model. The purpose of the U-CLX Model is to calculate a value with which to measure the level of U-Learning; it achieves this using the equation for the volume of a hypersphere. The volume equation allows us to find the value of U-Learning from the four dimensions or variables of the model. The minimum (0) and maximum (100) values of each dimension are defined; with these data, the minimum and maximum levels of ubiquitous learning can be obtained from the volume of the hypersphere in the equation.

## 5. Use of the U-CLX Model

This section details the evaluation environment, the general process, and the self-assessment tool.

### 5.1. U-CLX Model Evaluation Environment

The model proposes a way of assessing ubiquitous learning in a virtual higher education environment, where the institution, the program, and the course self-evaluate. The evaluation result indicates the level of U-Learning; that is, it indicates whether the level of ubiquitous learning is high, medium, or low. The measurement is obtained from the data and information of the institution, program, and course (see Appendix A). The U-CLX Model calculates the level of U-Learning, and the equation of the volume of the hypersphere is applied to calculate ubiquitous learning.

The Academic Managers, Learning Engineers, and Students are responsible for self-assessment in the application of the U-CLX Model, and the evaluation results measure the level of ubiquitous learning. With the information generated by the model, we can then propose actions to improve the institution's U-Learning level.

- University: At this level, the university has the data and information on the processes, experiences, procedures, policies, guidelines, etc. to allow ubiquitous learning in the institution to be assessed.
- Program: At this level, the institution has the data and information on the academic programs in which learning can take place, on the ubiquitous learning experiences of students in the program, and on how well-developed the ubiquitous learning processes and experiences are.

- Course: At this level, measurements of the strategies, means, processes, technologies, etc. of the courses assessed are available to the academic programs. These allow data on the ubiquitous learning experiences to be captured, managed, and measured to ensure that the learning objectives are achieved.

*5.2. U-CLX Model General Process*

The assessment process for measuring the level of ubiquitous learning begins with recognition of the roles of people in the institution—academic managers, learning engineers, and students—in applying the assessment survey of the U-CLX Model.

Self-evaluation is carried out at the university level to assess the levels of U-Learning in the institution. The questions cover policies, strategies, and other institutional elements to identify the institution's ubiquity level. Next, at the program level, self-evaluation is carried out to measure the levels of U-Learning in the program, where the pedagogical and technological components of the academic programs, processes, and experiences of ubiquitous learning programs are assessed. Finally, self-evaluation is conducted at the course level to measure the levels of U-Learning in the courses. The survey extracts data on the pedagogical components and elements in the ubiquitous course learning processes, as well as on the technology applied. The U-Learning measurement proposed in the U-CLX Model is calculated using the data obtained.

*5.3. U-CLX Model Self-Assessment*

The self-assessment surveys of the U-CLX Model were designed to cover the conceptualization as well as all the dimensions, components, and elements of the model. Furthermore, elements of the model are divided into items and subitems; the elements and items of each dimension allow U-Learning to be assessed in terms of the measuring unit UbiquoL. The rating scale of the U-CLX Model defines the U-Learning level of the institution.

The self-assessment survey is structured as follows: (a) a header with the evaluation title; (b) the role; and (c) the survey rating scale. In addition, the self-assessment survey has the following columns: Components and Elements.

Each element has a description, and there is an element subtotal at the end of each element. The subtotal of each component is shown, and, finally, a total average grade is given for each dimension.

All the dimensions of the U-CLX Model have the same weight in the calculations performed to define the ubiquitous learning level. The data for each dimension are averaged for the purposes of calculating its value. The level of each element is calculated from the average of all its headings. The levels of the dimensions are calculated by averaging each element and component.

The average per element is calculated as follows: for each element, the sum of the values of each variable is divided by four dimensions (Element/Sum of Dimension Ratings (4)).

The average per variable is calculated as follows: for each element, the sum of the values of each variable is divided by the number of elements (Sum of Dimension Ratings /Number of elements).

The average per component is calculated as follows: the totals of all the variables are added together and the sum is divided by the six elements; each component has six elements (sum of totals for each variable/number of elements of each component).

The mean total of the elements is calculated as follows: all the elements are summed and divided by the variables. The total elements are summed and divided by the number of elements (sum of the totals of each element/number of elements of each element).

The total average is calculated as follows: the totals of the two components in each variable are added together and the sum is divided by the two components (sum of the total of each variable per component/number of components (2)).

### 5.4. U-CLX Model Unit and Indicators

The U-CLX Model proposes a unit of measurement to measure ubiquitous learning, or U-Learning, called UbiquoL. UbiquoL is the relationship between the learning process and the characteristic of ubiquity. This unit allows for measuring the level of U-Learning in an institution [2].

We are still looking for a unit of measurement that quantifies the level of ubiquitous learning; for this reason, this measurement proposal is made in this model.

$$UbiquoL = Learning\ process/Ubiquity\ characteristic$$

In the U-CLX Model, we defined a scale from 1 to 5 with the unit of measurement "UbiquoL". This scale rates the level of ubiquitous learning, with one being the lowest and five the highest, as shown in Table 2. They are determined in order to facilitate the definition of U-Learning levels. The scale does not start at zero, because all institutions have ICT elements implemented.

**Table 2.** U-CLX Model Rating—units of measurement in UbiquoL [2].

| Unit of Measurement | Qualification | Description |
|---|---|---|
| **UbiquoL** | 1 | Low Level of U-Learning |
| | 2 | Medium-Low Level of U-Learning |
| | 3 | Medium Level of U-Learning |
| | 4 | Medium-High Level of U-Learning |
| | 5 | High Level of U-Learning |

In the U-CLX Model, a table has been defined with evaluation criteria to measure the level of ubiquitous learning. These criteria are related to the UbiquoL unit levels of the measurement rating scale, where one is the lowest level and five is the highest level. Table 3 shows, in percentage, what each rating represents, and classifies them into low, medium, and high.

**Table 3.** Levels, Criteria, and Percentage of the U-CLX Model [2].

| | Ubiquitous Learning Assessment Criteria | |
|---|---|---|
| **Level** | **Qualification** | **Percentage** |
| Low—L | 1 | 0–20% |
| | 2 | 21–40% |
| Medium—M | 3 | 41–60% |
| | 4 | 61–80% |
| High—H | 5 | 81–100% |

The U-CLX Model defines indicators that express the U-Learning results obtained with the model's mathematical application, which was proposed in Section 2. The indicators are defined as Low (L), Medium (M), and High (H): the lowest grade is (1), and the highest is (5). For example, an institution, a program, or a course is assessed as having a low level of learning (L) if their total grade is one or two, representing a total percentage of 0–40%. If the total score is (3) or (4), representing a percentage between 41–80%, the level is medium (M). Furthermore, if the total score is (5), a percentage of 81–100%, the level is rated as high (H). This rating has been constructed to define the level of U-Learning in an institution.

## 6. Validation and Application of the Proposal

The U-CLX Model validation process was divided into two parts: the conceptual and the mathematical validation.

The conceptual validation was based on the *Delphi* methodology, consulting different experts with doctorates in areas such as science, computing, education, mathematics, and

physics. We contacted them to participate in the expert evaluation because they work in Ubiquitous Learning or U-Learning, Information and Communication Technologies, or Applied Technologies in Education. Once the experts had been defined, the validation was carried out with the results obtained in the conceptual process of the U-CLX Model.

The validation was carried out using a panel of experts in mathematics and physics to validate the mathematical model. In this process, formulas and equations are essential to support the U-CLX Model. The expert panel approved the equation as a tool for calculating the level of U-Learning in the U-CLX Model.

### 6.1. Conceptual Validation

Conceptual validation of the U-CLX Model was carried out by experts using the Delphi methodology. Initially, a survey was designed with all the information on the U-CLX Model; then, we contacted different experts to invite them to participate in the evaluation by email. The twenty-one experts who agreed to participate in the evaluation received instructions on how to perform the evaluation, indicating that it was a blind evaluation in which they would not know who the other experts were until the end of the process, when the final results were presented. The next step was to send the information on the U-CLX Model to the experts, together with the evaluation survey. Once the experts had completed their evaluation, we reviewed, analyzed, and evaluated the answers, comments, observations, and assessments that they submitted.

We have made adjustments and changes to the model to improve it within this iteration. Once the changes had been completed, we designed a new evaluation survey based on the answers obtained in the first evaluation. We sent this new approach to the experts with the amended U-CLX Model for a second expert evaluation. Further changes and improvements were made to produce the final U-CLX Model (see Appendix C).

### 6.2. Mathematical Validation

Mathematical validation of the U-CLX Model was carried out by an expert panel of teachers of master's degree and doctoral courses with expertise in mathematics and physics. Five experts participated in the validation process. This expert panel conceptually evaluated the processes and mathematical equations used to calculate the U-CLX Model.

Under recommendation of a specialist in mathematics and physics, we decided to work with a hypersphere in Rn, specifically in R4, because the U-CLX Model has four dimensions. Use of a hypersphere allows equations to be developed for calculating the volume in four dimensions. We therefore proposed use of the volume of the hypersphere to calculate the level of ubiquitous learning according to the conditions and specific requirements.

The experts evaluated the process, equations, and the concept of calculating U-Learning by the formula of a hypersphere in R4. After reviewing the processes and validating the calculations, the five experts concluded that using these equations to measure the level of U-Learning is coherent, correct, and feasible.

### 6.3. U-CLX Model Application

The universities in which the U-CLX Model was applied are virtual distance universities with an excellent track record in virtual higher education at the local and international levels. They have people, processes, procedures, policies, pedagogies, technologies, and the elements for developing U-Learning processes, and they have the necessary U-Learning staff within their institutions to evaluate with the U-CLX Model.

The U-CLX Model was applied in two universities: Universidad Nacional Abierta y a Distancia—UNAD, a leader in virtual and distance education in Colombia; and Universidad Internacional de la Rioja from Spain, a leader in virtual education in Spain and Latin America. The two universities have developed e-learning processes and work in U-Learning.

The data were obtained from application of the U-CLX Model using the process designed for institutions. First, the self-evaluation survey was applied to each university's

academic manager, learning engineer, and students. With the data obtained, we measured the level of U-Learning of the institutions by applying the equation defined in the model. Data for each dimension were obtained through the self-evaluation surveys, and these data were used to calculate the level of U-Learning [2].

The surveys contained questions to enable each person to evaluate the institution according to his/her role. The questions were classified by the components and elements, and the person evaluated each element for the four dimensions: Time, Place, Medium, and Context, taking into account the levels, criteria, and percentages defined in Table 2. When all the respondents had completed the self-evaluation, the model calculated the value of each dimension. With the values of the dimensions, the model's equation was applied to measure ubiquitous learning and define the level of U-Learning in the institution. This was performed in both institutions. The results obtained were compared with the minimum and maximum levels of U-Learning defined in the model in order to assess the institutions' current state and propose improvement plans [2].

*6.4. Results*

The data obtained from evaluation by the U-CLX Model and associated learning analysis processes indicate that Universidad Internacional de la Rioja UNIR and Universidad Nacional Abierta y a Distancia UNAD have medium-high levels of ubiquitous learning processes. Comparison of the evaluations indicates that the former university obtained a higher score than the latter. This exercise aims to obtain feedback on the model's strengths and weaknesses. It is a statistically non-significant exercise.

The data indicate that the two institutions have strengths and should work on developing new paradigms to apply U-Learning. However, the technological component is weaker in both institutions, as they show lower grades with respect to the maximum value in the model. This indicates that they should build and improve their ubiquitous learning processes and experiences. One area that needs attention is the development and implementation of learning analysis processes, including new technologies to implement and manage U-Learning.

The dimension in which both institutions obtained the lowest scores was the Context, indicating that they should develop ubiquitous learning processes that allow for the inclusion of the natural world, virtual reality, augmented reality, and formal and informal education in ubiquitous learning (see Table 4).

**Table 4.** Level of Ubiquity of Institutions according to the U-CLX Model.

| Level of U-Learning | UbiquoL |
|---|---|
| Minimum Value | 79.0 |
| Value of the UNAD | 23,182.1 |
| Value of the UNIR | 29,150.8 |
| Maximum Value | 49,348.0 |

The radial chart shows the results obtained in each institution and the comparison with the minimum and maximum reference values of the U-CLX Model, as shown in Figure 7.

We conclude that the conceptual and mathematical bases of the U-CLX Model provide a valid way of measuring U-Learning. The data obtained from the model are indicative of the institution's current state, and they indicate how it can improve ubiquitous learning processes and experiences according to the components and elements assessed in the U-CLX Model (see Appendix B).

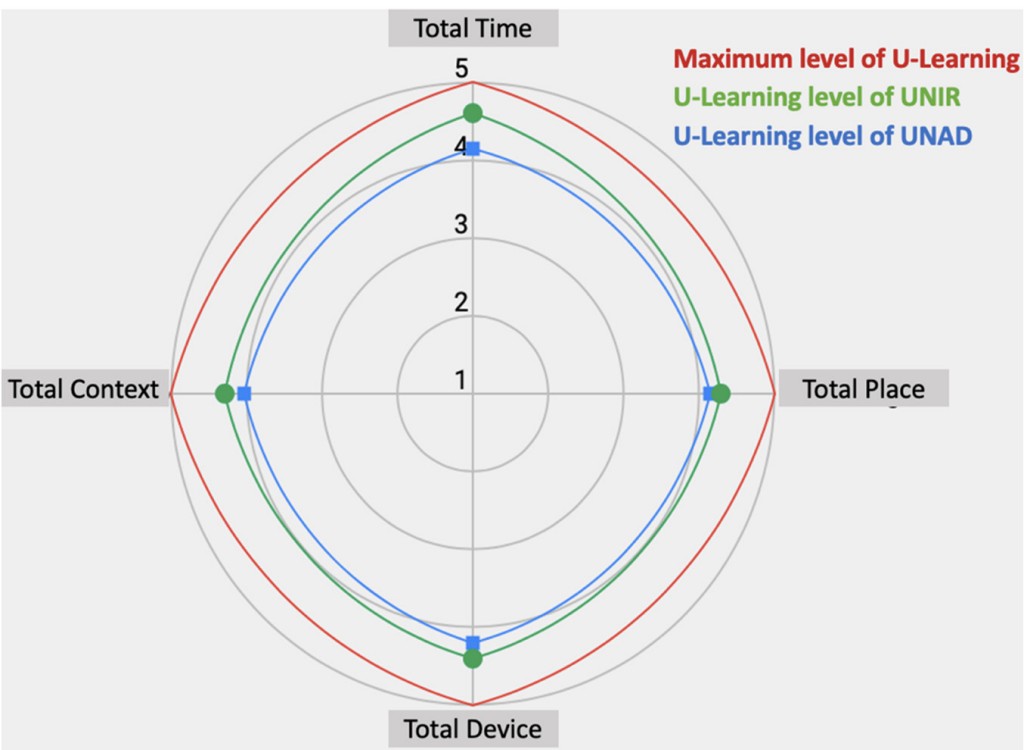

**Figure 7.** Comparison of the results [2].

## 7. Conclusions and Future Work

The U-Learning model, supported by the learning experiences and learning of the Virtual Higher Education Connective, proposes a way of measuring ubiquitous learning. We describe the concepts, characterization, and mathematical foundation which allow the level of U-Learning in virtual higher education to be assessed. This information will allow institutions to improve their educational processes and technologies for ubiquitous learning. It allows us to know the current state of U-Learning in an institution, and what is needed to make it a reality.

The present proposal is the result of several iterations. As we found results, these were published, and they strengthened our initial hypotheses. We highlighted mainly three. In the first article, we provided evidence of the common elements of selected models and their mechanisms of integration [7]. In the second, we exposed theoretical bases on connective learning and xAPI to develop the u-Learning framework [9]. Finally, we explored the possibility of integrating the xAPI standard to manage the learning experiences from ubiquitous teaching–learning processes. The proposal presents two components: pedagogical and technological [42].

The U-CLX Model proposes a U-Learning ecosystem composed of four dimensions: Time, Place, Medium, and Context. This ecosystem contains a Pedagogical Component based on ubiquitous learning and a Technological Component based on ubiquitous computing. Each component includes six elements. The elements of the Pedagogical Component are Learning Paradigms, Learning Scenarios, Academic Levels, Knowledge Domains, Physical Characteristics, and Effects; the elements of the Technological Component are Learning Experiences, Technologies, Learning Analytics, Functionality, Devices, and Management. Some of these 12 elements contain items and subitems.

The mathematical basis of the U-CLX Model is based on the concepts of lines, line segments, hyperplanes, convex sets, and hyperspheres in four dimensions, R4; the four dimensions become the model variables. Equation (7) allows the level of ubiquitous learning to be calculated through the volume calculation in a hypersphere, R4. The positive values of each variable between 0 and 100 (20) are used; the data for each variable are

obtained from the self-assessment surveys. With these data, the volume is calculated to measure the level of ubiquitous learning.

The results indicate that the U-CLX Model performs the ubiquitous learning level measurement considering the model conceptualization, dimensions, components, elements and items, the evaluation processes, and the roles of the individuals. In other words, the U-CLX measures the level of ubiquitous learning of a virtual higher education institution, presenting the current state of U-Learning and how it could be improved.

It is essential to clarify that other models measure learning, online education, blended learning, the nature of education, and the forms of knowledge. However, our previous literature reviews didn't find any initiative that measured ubiquitous learning in virtual education institutions. Only the TAG model measured the institutions' ubiquity level. Even when the U-CLX Model was developed, no models or methodologies were found to measure U-Learning in virtual education institutions.

Future work for the U-CLX Model might be focused on including new dimensions, and allowing measurement of other variables; this could be performed by using a hypersphere with more dimensions. Another important area for research would be how to carry out self-evaluation in real-time, taking the data automatically and having a permanent display of the results of the dimensions; this will allow constant monitoring of progress. Finally, it is vital in the future to implement AI in the development of U-Learning evaluation; this will allow better results to be obtained from the data submitted in the evaluation surveys.

**Author Contributions:** Conceptualization, G.M.R.V. and C.A.C.; methodology, G.M.R.V.; validation, G.M.R.V., C.A.C. and J.D.; formal analysis, G.M.R.V.; investigation, G.M.R.V., C.A.C. and J.D.; writing—original draft preparation, G.M.R.V. and J.D.; writing—review and editing, J.D.; visualization, J.D.; supervision, C.A.C.; funding acquisition, J.D. All authors have read and agreed to the published version of the manuscript.

**Funding:** This research was funded by Dirección de Investigación, and project DI21-0016, Universidad de La Frontera, Temuco, Chile.

**Institutional Review Board Statement:** Not applicable.

**Informed Consent Statement:** Not applicable.

**Data Availability Statement:** Not applicable.

**Acknowledgments:** We thank the experiment participants, specialists, and technicians who supported this initiative. Special thanks to the members of the User Experience & Game Design (UXGD)-UFRO and HCI-COLLAB research groups.

**Conflicts of Interest:** The authors declare no conflict of interest.

### Appendix A

Main dimensions of the U-CLX model (accessed on 11 November 2022)

- https://docs.google.com/spreadsheets/d/1kSCJ_NdtnDkcWPoKZlZs3IlPlNrsOJzGVlXmV_nXANk/edit?usp=sharing

### Appendix B

UNAD-UNIR Learning Analytics Dashboard vs UNAD-UNIR (accessed on 11 November 2022)

- https://datastudio.google.com/open/1ofiR0etDw3ic_xSZk2KnSQPF72V8IqcV

### Appendix C

Instrument: Expert evaluation using Delphi methodology (accessed on 11 November 2022)

- https://forms.gle/YAFsmmv9TnbhHZ3WA
- https://forms.gle/eFmmE1DkgBzLt2Wj8

Results: Expert evaluation using Delphi methodology (accessed on 11 November 2022)

- https://docs.google.com/spreadsheets/d/10uYfSvx0E-Y_TKtjXjL6AlCtwqaACSYkxBJpdn_0XCs/edit?usp=sharing
- https://docs.google.com/spreadsheets/d/1Z7Kz0ueqt9Dwzpl64OGjUpHbUVD82KDLjUGjENX--rE/edit?usp=sharing

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
