# Peer review of "A Model to Measure U-Learning in Virtual Higher Education: U-CLX"

_applsci, doi:10.3390/app13021091_

Round 1

Reviewer 1 Report

The manuscript is well-written and, in a general way, well-structured. Maybe the authors could rethink the number of sections since it has ten.

The title and abstract are engaging. In the abstract, I suggest clarifying the research problem and the contributions of the work. Related to the research problem, an important point that the manuscript could highlight is why the model is needed, what challenges the educational institutions are facing, and what can benefit from the model created.

In the introduction, some elements are missing, such as the research problem, the research question, and the justification for creating the model. The last paragraph of the introduction could be divided in two, and a new paragraph could provide the details regarding the article structure.

The manuscript presents two systematic literature reviews that were carried out to get insights to construct the model. The manuscript could present related work found and discuss the main contributions of their model based on this literature.

Some details regarding the model are missing, for instance, the details of what is measured in each of the four dimensions of the model. This information can improve the evaluation of the model.

Also, the first Table 2 (there are two Table 2) presents the unit of measure, but it is unclear what a low or a high level means. It is important to give this information. Maybe the manuscript could provide a sample to improve the understanding of the model.

The survey applied to the two Universities to evaluate the model could be provided as an appendix and could help in the understanding of the items mentioned above.

The Tables legend must be reviewed, the numbers are not correct.

Author Response

Dear Reviewer,
We appreciate each of your comments. They have been very rewarding for our team and have helped refine our current proposal.

We have responded to each of your comments.

Reviewer 2 Report

**Updated 10.16  English version is in attached file -

Premetto innanzitutto che il lavoro appare molto vicino - a volte è solo una traduzione - ad una tesi di dottorato di uno degli autori del 2019: "Model U-Learning Soportado por las Experiencias de Aprendizaje y el Aprendizaje Conectivo per la Educación Superior Virtual - U-CLX ". In ogni caso, il lavoro è scritto molto male. Ecco solo alcune considerazioni illustrative:

In linea di principio e considerando l'ambizione dell'articolo, trovo scomodo presentare praticamente un sondaggio all'interno dell'opera che ha altri obiettivi. Ho avuto la sensazione che la prima parte dell'articolo si riferisse ad un altro articolo perché non introduce gli obiettivi del lavoro presentato.

L'attuale articolazione del documento con un'impostazione che vede all'interno della doc l'approccio tipico di un sondaggio, non serve, appesantisce il lavoro, distrae dagli obiettivi e non serve ai fini del lavoro.

Consiglio di eliminarlo e sintetizzarlo in modo appropriato sullo sfondo.

Inoltre, è molto generico e non è molto focalizzato sull'obiettivo dell'articolo.

Finally, some considerations present such as "However, the systematic review found few works on the implementation of ICT and educational processes", are absolutely not acceptable, as if a distance learning expert performs a search he has exactly the problem opposite to. the problem is precisely in poor research and in the misuse of keywords.

In fact, it should be noted that in the words used to carry out the search I see terms that are too general and above all words such as evaluation metrics, learning evaluation, etc. are missing. So this whole part is of no help and in my opinion heavily invalidates the proposed work due to its generic nature.

The background paragraph is disappointing, in practice a review of some - however limited definitions - Here the reader expects to know the state of the art in the field of evaluation and metrics adopted in the field of distance learning, in particular the u -learning with a good vision on the evaluation models and metrics of other forms of distance learning useful to then introduce and define one's own model and possibly compare it with other existing ones. The part of the state of the art and background necessary to frame the article is practically absent even in the presence of a certainly demanding research work carried out but not very useful for the final objective.

To be very clear it is not clear whether the authors have references to: The approaches according to Stame, Stufflebeam and Webster, or the approaches according to Madaus, Haney, Kreitzer, as well as the models in depth: Scriven's goal-free evaluation; Stufflebeam's CIPP Model; Stake's responsive evaluation; Eisner's connoisseurship evaluation… and above all to the most recent and modern evaluation models produced for blended and fully online training. I believe these are starting points that cannot be eliminated in a background suitable for the proposed objective, in addition of course to the model considered on which they also dwell excessively.

In practice, if the authors want to convince and well support their proposal, they cannot limit themselves to stating Currently, there are models, methodologies and frameworks that have been tested to measure the level of ubiquitous learning of educational institutions, such as the TAG model [26]. Learning processes, technologies and other areas have been examined by Bomsdorf [15], Kwon [27] and Poslad [28], but so far there is no model to measure U-Learning in virtual higher education, without passing reviews existing valuation models.

I do not enter into the merits of the proposed model, because the current presentation has too many defects. However I observe that in terms of originality I only see a dimensional extension of the original model, which could also be of some interest but not in the current formulation.

Quello che vorrei raccomandare agli autori, partendo dal lavoro di tesi di dottorato, è che non è sufficiente una sintesi della stessa, ma un miglioramento e una focalizzazione sul metodo di valutazione.

In ogni caso, la forma attuale del lavoro, in termini di organizzazione e descrizione, non è in una forma accettabile per una rivista scientifica.

Author Response

(The authors gave the same response as above.)

Reviewer 3 Report

The subject of this paper is important. U-Learning is a new and complicated learning paradigm incorporating various dimensions that should be taken into account in order to be measured and evaluated. So, a model for such an evaluation can be an important contribution to the relative literature. 

However, the particular aim of this paper are not clearly presented. What aspects of U-Learning  are measured by the proposed framework? For example, level of adoption, pedagogic effectiveness, technical quality, etc.

In section 4, the twelve elements of U-Learning are presented together with the four dimensions and four components of the model.  It is not clear how these elements, dimensions and components are related. Particularly the quantification of the four  dimensions should be explained, since these dimensions are calculated in the mathematical part of the model. For example, if there is a questionnaire or rubric that quantify the proposed dimensions, they should somehow described in the paper.

The mathematical part of the model needs to be revised. Since the variables cannot be negative (p. 11), the whole hyperspace is not defined in the model and the calculated (maximum) volume is the one sixteenth of the volume of the hypersphere. Furthermore, it is not clear what is the meaning of the volume of the hypersphere that is used as a measure in the model. There are some problems with mathematical symbols, for example, the symbol 'p' on page 12 possibly corresponds to the symbol Greek rho on equations 3-6 on the same page. On page 13, eq. 11, there is a pi squared that is equal to the first power of pi on the same equation.

The validation of the model does not provide evidence for the mathematical model that is proposed as a metric of U-Learning. It is not justified why the particular measure was used, instead of another. For example, in the proposed mathematical formula, all four dimensions  of U-learning are considered of equal importance/ contribution to the final metric and this should be justified in the paper.

Sections 5-10 that report on the evaluation of the proposed model should be merged in one single section with multiple subsections.

Author Response

(The authors gave the same response as above.)

Round 2

Reviewer 1 Report

The research problem and research question could also be included in the abstract. Also, it is important to review the writing. For instance, the introduction's last phrase mentions section 11, but the manuscript has only seven sections. Furthermore, some new paragraphs have only one phrase, it is interesting to keep the same pattern used in the rest of the text. Finally, the first Appendix is inaccessible.

Author Response

Dear reviewer,
Thank you for your feedback.

I am attaching our response to your comments.
Kind regards.

Reviewer 2 Report

The authors made some adjustments.

The changes introduced are partial adjustments and do not change my overall opinion on the work from a scientific point of view.

I remain convinced that the pseudo-survey is absolutely to be eliminated and does not provide any contribution on the validation of the article despite what the authors say.

This article deals with a potentially very interesting topic and proposes the extension of a model treated in other articles and formulated in a doctoral thesis.

The principal critical point is that the current formulation and articulation of the article does not allow a satisfactory evaluation from a scientific point of view.

In my opinion the authors should rewrite the article.

I suggest you:

- review the introduction where it appears necessary to make the reader understand the transition between e-learning and u-learning and the need for a formal strategic measurement model for the organization that adopts the U-Learning modality. This Goal must be clear. Furthermore, it is necessary to provide a clear understanding, of this aspect, respect to existing u-learning measurement models, including partial experiences of evaluating u-learning actions in the literature. A cultured and detailed approach in this area is crucial and necessary for a correct insertion of the work.

- formulate a correct background and modern state of the art with respect to models for measuring U-Learning in higher virtual education. I invite the authors to reflect on the fact that in the context of the backgound the paragraph 3.6 "U-Learning Measurement models" which is the heart of their article is reduced to 5 lines, without any discussion and classification of the same. Be clear and transparent about the genesis of the proposed model. In that context, frame their previous articles (clarifying the differences and advances) and thesis work. It is necessary to provide a clear vision of the current state of the art at world level. To this end, see a simple search using "U-Learning measurement models" as a search key from 2010 or 2015 onwards which produces many interesting, albeit partial, results. This relevant aspect in every scientific article unfortunately in the current formulation is in fact not present and summarized with the phrase that there are no models, while it is evident and easily verifiable that there are partial solutions and other experiences in this sense in the field of U-Learning.

- Present their model taking care not to get lost in elementary aspects of geometry. I point out once again that in the current version it is not clearly understood in what the model presented in this article differs from other articles by the authors. Has the model already been presented or not? If it has already been presented, I think it is necessary to better understand the meaning of the new article.

- Carry out an evaluation and above all a comparison with other U-Learning measurement models or at least experiences present in the U-Learning literature. Here a formal, qualitative-quantitative and convincing discussion is needed.

- Present the trials in a formal and scientifically valid way and with more elements.

While appreciating some of the changes introduced respect the precedent version, I repeat that the current wording is not adequate.

Author Response

(The authors gave the same response as above.)

Reviewer 3 Report

There is still a problem with the powers of pi in the mathematical formulae on page 15, lines 581 and 583  that should be corrected. On these formulae the first power of pi is equaled to the second power of pi, which is apparently a mistake.

Author Response

(The authors gave the same response as above.)

Round 3

Reviewer 2 Report

The current form is acceptable and certainly makes the objectives of the work clearer. I also believe that there is significant room for improvement both for the model and for its possible applications.